# ROUTOO: LEARNING TO ROUTE TO LARGE LANGUAGE MODELS EFFECTIVELY

## ABSTRACT

LLMs with superior response quality—particularly larger or closed-source models—often come with higher inference costs, making their deployment inefficient and costly. Meanwhile, developing foundational LLMs from scratch is becoming increasingly resource-intensive and impractical for many applications. To address the challenge of balancing quality and cost, we introduce Routoo, an architecture designed to optimize the selection of LLMs for specific prompts based on performance, cost, and efficiency. Routoo provides controllability over the trade-off between inference cost and quality, enabling significant reductions in inference costs for a given quality requirement. Routoo comprises two key components: a performance predictor and cost-aware selector. The performance predictor is a lightweight LLM that estimates the expected performance of various underlying LLMs on a given prompt without executing them. The cost-aware selector module then selects the most suitable model based on these predictions and constraints such as cost and latency, significantly reducing inference costs for the same quality. We evaluated Routoo using the MMLU benchmark across 57 domains employing open-source models. Our results show that Routoo matches the performance of the Mixtral 8x7b model while reducing inference costs by one-third. Additionally, by allowing increased costs, Routoo surpasses Mixtral's accuracy by over 5% at equivalent costs, achieving an accuracy of 75.9%. When integrating GPT4 into our model pool, Routoo nearly matches GPT4's performance at half the cost and exceeds it with a 25% cost reduction. These outcomes highlight Routoo's potential to significantly reduce inference costs without compromising quality, and even to establish new state-of-the-art results by leveraging the collective capabilities of multiple LLMs.

## 1 INTRODUCTION

LLMs have achieved remarkable success across a wide range of natural language processing tasks. However, this success comes with a significant trade-off between performance and cost: larger models generally offer better response quality but incur higher inference costs than their smaller counterparts. This increased inference cost poses challenges for deploying LLMs in real-world applications where computational resources, latency, and cost-efficiency are critical considerations.

At the same time, developing foundational LLMs (OpenAI et al., 2024; Dubey et al., 2024; Yang et al., 2024; DeepSeek-AI et al., 2024; Jiang et al., 2024; 2023a; Touvron et al., 2023) from scratch is becoming increasingly capital-intensive, requiring vast computational resources and extensive, high-quality data (Minaee et al., 2024). As the field approaches the limits of network size and data capacity, the improvements gained from training ever-larger models are diminishing (Udandarao et al., 2024). This situation underscores the need for alternative approaches that can achieve high performance without incurring prohibitive development and inference costs.

Moreover, many practical tasks do not necessitate the intricate reasoning capabilities of the largest models like GPT-4; instead, they can be efficiently handled by smaller, more cost-effective models (Zaharia et al., 2024; Ding et al., 2024; Shnitzer et al., 2023; Chen et al., 2023). The capabilities of existing LLMs appear to be complementary to a significant degree. For example, on the MMLU benchmark (Hendrycks et al., 2021), selecting the optimal open-source model for each question could hypothetically yield an accuracy of 97.5% at a computational cost similar to that of a 13-

billion-parameter model (see Appendix A for details). In contrast, GPT-4 (OpenAI et al., 2024) achieves an accuracy of 86.4%, while Mixtral 8×7b (Jiang et al., 2024), one of the leading open-source models,[1] reaches 70%. These observations suggest that integrating the knowledge of multiple LLMs could lead to new state-of-the-art models without the need to train from scratch while signifcaly reducing the inference cost. However, effectively leveraging the vast and rapidly growing ecosystem of LLMs poses significant challenges. With approximately 450,000 models available on Hugging Face,[2] keeping track of the latest advancements and integrating them efficiently is a non-trivial task. Traditional approaches like Mixture of Experts (MoE) models (Abdin et al., 2024; Lieber et al., 2024; Jiang et al., 2024; Fedus et al., 2021; Shazeer et al., 2017) are limited by the need to load all expert parameters onto a single high-end machine, hindering scalability and flexibility. There is a need for an architecture that can dynamically and efficiently leverage multiple existing LLMs to optimize performance while controlling inference costs.

To address these challenges, we propose Routoo , an architecture designed to optimize the selection of LLMs for specific prompts based on performance, cost, and efficiency. Routoo provides controllability over the trade-off between inference cost and quality, enabling significant reductions in inference costs for a given quality requirement. By intelligently leveraging a universe of trained LLMs, Routoo addresses both the inference cost problem and the development cost of building LLMs from scratch, creating a composite high-performance model without the need for extensive retraining.

Routoo comprises two key components: a performance predictor and cost-aware selector. The performance predictor is a lightweight LLM that estimates the expected performance of various underlying LLMs on a given query without executing them. Based on these predictions and constraints such as cost and latency, the cost-aware selector module selects the most suitable model. For instance, when a task is predicted to be performed nearly equally well by a smaller model or a larger, more expensive model, Routoo will opt for the smaller model when speed and cost-efficiency are prioritized. This approach ensures optimal resource utilization without compromising on quality.

Our architecture marks a significant departure from traditional MoE. While MoE relies on gating over various expert sub-networks within each layer to predict the next token, it requires all expert parameters to be loaded onto a single, high-end machine. This limitation hinders scalability in the number of experts. In contrast, each 'expert' within our system operates independently and can be hosted on different machines, potentially utilizing a different neural network architecture. This flexibility enables Routoo to incorporate a vast array of experts, ranging from specialized domain models to general-purpose ones, and to scale to a large number of experts without the limitations imposed by MoE models.

We evaluate our Routoo on MMLU benchmark (Hendrycks et al., 2021). We show that it achieves competitive performance with Mixtral 8x7b (Jiang et al., 2024) while only consuming two-thirds of its inference cost. Increasing the cost budget allows Routoo to outperform Mixtral by 5% at the same cost level, reaching an accuracy of 75.9%. By adding GPT4 (OpenAI et al., 2024) as one of the underlying experts, our Routoo achieves competitive performance with GPT4 at half the cost and exceeds it with a 25% cost reduction. These outcomes highlight Routoo's potential to significantly reduce inference costs without compromising quality, and even to establish new state-of-the-art results by leveraging the collective capabilities of multiple LLMs. By providing controllability over cost and quality trade-offs, Routoo offers an efficient approach to high-performance language modeling without the need for expensive model training from scratch.

To summarize, our contributions are:

- We propose Routoo, an LLM-based system that intelligently identifies the best-performing LLM for a given query while considering constraints such as cost and latency, effectively integrating the knowledge of multiple LLMs to create a high-performance model without the need for training from scratch.

- We evaluate our architecture on MMLU benchmark, and significantly outperform Mixtral 8x7b by 5% with similar inference cost. Also, our Routoo achieves competitive perfor-

---

[1]As of the time of writing this paper.

[2]https://huggingface.co

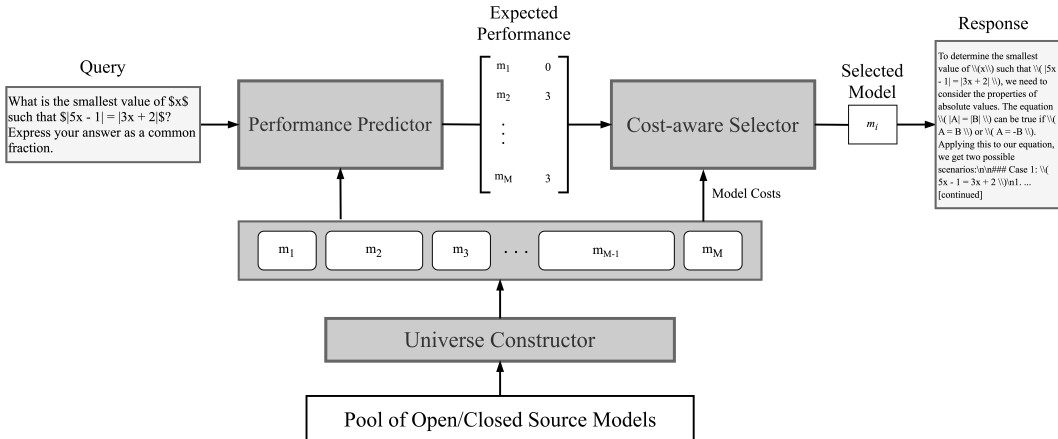

Figure 1: Routoo architecture has three main components: a performance predictor, cost-aware selector, and universe constructor. The universe constructor identifies the optimal set of complementary models from all available models. The performance predictor predicts the correctness of experts for a specified query, and the cost-aware selector chooses the underlying model by considering the cost and efficiency of each model.

mance with GPT4 at half the inference cost and surpasses it by reducing the inference cost by 25%.

## 2 RELATED WORK

**Mixture-of-Experts.** MoE architecture includes a gating mechanism to integrate various expert sub-networks within each layer, guiding the prediction of the next token (Shazeer et al., 2017). This approach is used in recent foundational models, including Mixtral 8x7b (Jiang et al., 2024). Sukhbaatar et al. (2024) fine-tuned LLaMa 7b (Touvron et al., 2023) on four different domains to create domain-specific expert LLMs. They then proposed combining the FFNs of these expert LLMs to construct an MoE. Tang et al. (2024) proposed merging most parameters while scaling up MLP layrers of Transformer into a weight-ensembling MoE module that dynamically integrates shared and task-specific knowledge based on the input. Wang et al. (2024) introduces a fusion gate at each Transformer layer to generate weights for a weighted average of outputs from a set of pre-trained LoRAs.

**Model Selection.** Previous works in selecting the best LLMs mainly focus on identifying the one that generates the most optimal output for a given input. Liu & Liu (2021); Ravaut et al. (2022) proposed specialized scoring or re-ranking models that can be used for the generation tasks (summarisation, here). Speculative decoding (Kim et al., 2023; Leviathan et al., 2023) accelerates the decoding of expensive models by using small, efficient decoders for the 'easy' steps. This approach can complement routing methods. ROUTERBENCH (Hu et al., 2024) proposed a benchmark for LLM routing task. akota et al. (2024) used meta-modeling approach to assign the appropriate model to each query while considering the cost constraint. LLM-BLENDER (Jiang et al., 2023b) is an ensembling framework to reach better performance by mixing the results of LLMs with a ranking module, followed by a generation module to generate from top candidates of the ranker. Frugal-GPT (Chen et al., 2023) executes experts sequentially until an expert reaches the acceptable generation performance. ZOOTER (Lu et al., 2024) proposed a reward-guided routing approach that distills rewards from training queries to train a routing function. HybridLLM (Ding et al., 2024) and RouteLLM (Ong et al., 2024) propose a routing approach to direct queries to the suitable expert, utilizing one strong and one weak LLMs.

Different from previous work, our Routoo identifies the most suitable expert without executing the inference of underlying LLMs. Additionally, our approach can be efficiently generalized to scenarios involving many underlying LLMs (not necessarily a tuple of weak and strong LLMs). This allows Routoo model to outperform even the best closed-source models, including GPT-4 (OpenAI et al.,

2024), by routing nearly half of the queries to open-source models (further details are provided in Section 4). Finally, Routoo identifies the optimal set of underlying models automatically (introduced as Universe Constructor in Section 3.4), a task that was not explored in any previous work, which relied on manual selection.

# 3 ARCHITECTURE

## 3.1 PROBLEM FORMULATION

Given a set of $M$ models $m_1, m_2, \ldots, m_M$ and a set of $N$ queries $q_1, q_2, \ldots, q_N$,[3] our goal is to assign the most cost-effective model $m_i$ to each query $q_j$ that can accurately answer the query. Correctness and cost scores of each query-model pair are calculated as:

$$s_{m_i,q_j} = \text{Eval}(m_i(q_j))$$
$$c_{m_i,q_j} = \text{Cost}(m_i, q_j)$$

where $\text{Eval}(.)$ computes the correctness score given the generated response $m_i(q_j)$ of model $m_i$ for query $q_j$. $s_{m_i,q_j}$ is an integer that ranges from 0 to $K-1$, indicating the correctness of the model's response, where 0 is unacceptable, and $K-1$ is optimal. The cost of executing model $m_i$ for query $q_j$ is calculated by $\text{Cost}(.)$ function.

The objective is to maximize the total correctness scores across all queries while keeping the total cost within a budget $B$:

$$\max_{\pi} \sum_{j=1}^{N} s_{\pi(q_j),q_j}$$

$$\text{s.t.} \sum_{j=1}^{N} c_{\pi(q_j),q_j} \leq B$$

Here, $\pi(.)$ is the function that assigns each query to a model from the set of $M$ models, while considering both cost and correctness. In this formulation, the cost can be multi-faceted, encompassing aspects like computational cost, speed, etc. However, for simplicity, we consider a singular budget constraint $B$ in this context. To solve this without exhaustively testing every model on every query—a prohibitive approach—we divide the problem into two manageable parts:

1. **Performance Predictor.** This component approximates the correctness score $s_{m_i,q_j}$ for each model on each query. Given a model $m_i$ and a query $q_j$, it estimates the model's performance without generating the response for the query, thereby significantly reducing the inference cost.

2. **Cost-Aware Selector.** This module selects the best model for each query, balancing accuracy and cost-effectiveness by using correctness estimations of performance predictor.

These two components work together to dynamically and efficiently allocate models to queries. Additionally, given the vast array of LLMs available for text generation,[4] and considering the practical constraints on the number of models that can be actively served, we introduce **Universe Constructor**, which builds an initial complementary universe of models $(m_1, m_2, \ldots, m_M)$ from a set of all available models. This universe aims to maximize performance by ensuring that the selected models are complementary to each other.

All three components are illustrated in Figure 1. In the following sections, we will delve deeper into each component.

---

[3] Section 3.4 will provide a detailed explanation of how we build the set of models from the pool of all available models.

[4] According to the Huggingface platform (https://huggingface.co), there are nearly 47,000 models available for the text generation task.

## 3.2 PERFORMANCE PREDICTOR

The performance predictor is a lightweight LLM designed to estimate the effectiveness of each underlying LLM for a given query. It estimates the output of evaluation function ($s_{m_i,q_j}$, introduced in Section 3.1) without executing model $m_i$ on query $q_j$. This prediction is formulated as:

$$\hat{s}_{m_i,q_j} = \text{Pred}(m_i, q_j)$$

where $\text{Pred}(m_i, q_j)$ is the predictive model. Similar to $s_{m_i,q_j}$, $\hat{s}_{m_i,q_j}$ is an integer that ranges from

---

**Algorithm 1:** Greedy algorithm for the optimization of universe construction.

---

**Result:** Find the set U of size k that maximizes $S(U)$ according to equation (2)
Initialize an empty set $U = \{\}$, and set max budget to M
Set budget to 0
**while** *budget < M* **do**
    1. Identify the model $m_i^*$ that when added to $U$ forming $U^*$ which maximizes $S(U^*)$
    2. Update $U$ by adding $m_i^*$
    3. Increment budget by 1
    **if** *No further improvement in $S(U)$* **then**
        | break;
    **end**
**end**

---

0 to $K-1$, 0 indicates an unacceptable result and $K-1$ represents the optimal outcome.

The process of $\text{Pred}(m_i, q_j)$ is as follows:

$$\begin{cases} h_{q_j} = \text{Enc}(q_j) \\ h_{m_i} = \text{Emb}(m_i) \\ \hat{s}_{m_i,q_j} = \text{Linear}(h_{q_j} - h_{m_i}) \end{cases}$$

where $\text{Enc}(.)$ is the encoder of the input query. Inspired by Radford et al. (2019), we use a decoder-only LLM as the query encoder and extract the embedding of the last token as the representation of the input query ($h_{q_j} \in \mathbb{R}^h$). $\text{Emb}(.)$ is an embedding layer, where each embedding is assigned to a specific model. As a result, $h_{e_i} \in \mathbb{R}^h$ is the embedding representation of model $e_i$. $h$ is the hidden representation of the decoder-only model used. $\text{Linear}(.)$ function is a projection matrix ($\mathbb{R}^{h \times K}$), computing the estimation score $\hat{s}_{m_i,q_j}$.

The model is trained by minimising the cross-entropy loss function (Good, 1952) of $s_{m_i,q_j}$ and $\hat{s}_{m_i,q_j}$ over all $M$ models, and $N$ queries as:

$$\min \frac{1}{N \times M} \sum_{i=1}^{M} \sum_{j=1}^{N} \text{CE}(s_{m_i,q_j}, \hat{s}_{m_i,q_j}) \tag{1}$$

where $\text{CE}(.)$ is the cross-entropy loss.

During the inference time, $\text{argmax}(.)$ function is applied for a given model and query.

## 3.3 COST-AWARE SELECTOR

The second phase of our architecture involves the selection step, where the estimated scores from the performance predictor are used to determine the optimal assignment of models to queries. Our objective is to maximize the overall effectiveness of the responses within a given budget constraint. The optimization problem is formulated as follows:

$$\max_{\pi} \sum_{j=1}^{N} \hat{s}_{\pi(q_j),q_j}$$

$$\text{s.t.} \sum_{j=1}^{N} c_{\pi(q_j),q_j} \leq B$$

| Model | Accuracy | Cost ($/1M tok) |
|---|---|---|
| LLaMa2 7b | 45.3 | 0.2 |
| Mistral 7b | 64.2 | 0.2 |
| LLaMa2 13b | 54.8 | 0.26 |
| Mixtral 8x7b | 70.6 | 0.6 |
| LLaMa2 70b | 69.9 | 0.9 |
| KNN (open-source) | 69.5 | 0.6 |
| Routoo (open-source) | 75.87 | 0.6 |
| GPT3.5 | 70.0 | 1.5 |
| GPT4-turbo | 86.4 | 20 |
| KNN (mix) | 79.1 | 10.2 |
| Routoo (mix) | 84.9 | 10.2 |

Table 1: Performance and cost of running LLMs on MMLU benchmark. Accuracy is calculated based on OpenLLM Leaderboard setting Beeching et al. (2023).

where $\pi(.)$ determines the model assignments and $B$ represents the predefined budget constraint.

We propose a greedy algorithm to approximate this optimization. Noting the challenges of accurately estimating the exact cost of a model for a specific query, instead, we consider the average cost $c_i$ for a model $m_i$ responding to an average length query and response. The algorithm includes the following steps:

1. **Performance-to-Cost Ratio:** For each query $q_j$ and model $m_i$, calculate the performance-to-cost ratio. Let $\hat{s}_{m_i,q_j}$ be the estimated correctness score of model $m_i$ for query $q_j$, and $c_i$ be the cost for model $m_i$. The ratio is calculated as:

$$r_{m_i,q_j} = \frac{\hat{s}_{m_i,q_j}}{c_i^\alpha}$$

   where $\alpha$ is a parameter that adjusts the emphasis on cost. A higher $\alpha$ value increases the weight on cost efficiency, thereby favoring cheaper models and preserving more of the budget for subsequent assignments.

2. **Sorting and Assignment:** For each query, sort the models by their performance-to-cost ratio in descending order. Assign the query to the model with the highest ratio that remains within the available budget.

3. **Budget Management:** Monitor the accumulated cost. If selecting a model would exceed the budget, move to the next best model in terms of the ratio.

This approach efficiently balances the trade-off between performance and cost, effectively identifying the most suitable model for each query while adhering to budget constraints.

### 3.4 UNIVERSE CONSTRUCTOR

With the abundance of models and the practical limitations on hosting several models, we introduce an optimization approach to build a complementary subset of models $(m_1, m_2, \ldots, m_M)$. The goal is to select models that are complementary to achieve the highest performance, given a predefined limitation on number of serving models.

The objective is to select a subset of models that collectively provide the best coverage and performance across a set of queries. Formally, given a big pool of models $\Omega : (m_1, m_2, \ldots, m_{\hat{M}})$, a set of queries $Q : (q_1, q_2, \ldots, q_L)$, and correctness scores of the models for these queries $s_{m_i,q_j}$, we seek to find the optimal subset of models, that maximizes the following optimization problem:

$$\max_{U \subseteq \Omega, |U|=M} S(U) = \frac{1}{L} \sum_{j=1}^{L} \max_{i \in U} s_{m_i,q_j} \tag{2}$$

where $M$ is the desired number of serving models, and $U$ is a subset of selected models from $\Omega$. The function $S(U)$ represents the highest score achievable by the set $U$, quantifying the com-

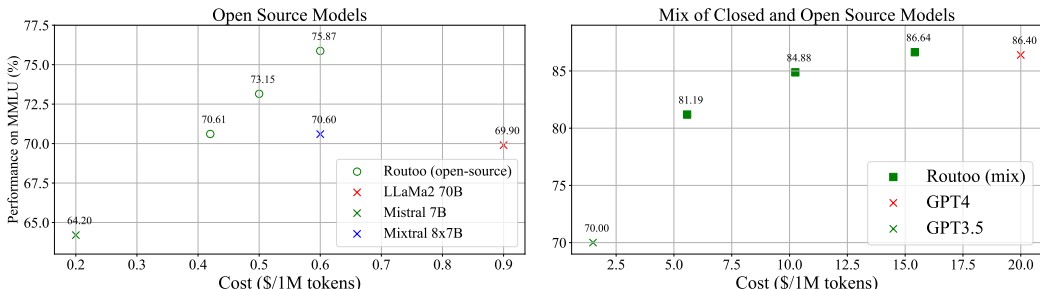

Figure 2: The performance of different Routoo models and baselines on MMLU benchmark, given different budget limitations.

bined performance of the selected models. The optimal subset $S(U^*)$ is used as the set of models $(m_1, m_2, \ldots, m_M)$ in the optimization of Section 3.1.

Given the submodular nature of the maximum operation in our objective function (Equation 2), a greedy algorithm (as illustrated in Algorithm 1) can be employed to find an approximate solution with acceptable error bounds (Krause & Golovin, 2014). This method is particularly effective for large-scale problems where exact optimization is computationally prohibitive.

## 4 RESULTS AND DISCUSSION

### 4.1 EXPERIMENT SETTING

**Evaluation Setting.** The evaluation of baselines and Routoo models is conducted using MMLU benchmark (Hendrycks et al., 2021), which comprises multiple-choice questions across 57 diverse domains, such as mathematics, law, computer science, biology, and US history. To be compatible with OpenLLM Leaderboard (Beeching et al., 2023), we use Eleuther AI Harness (Gao et al., 2021) to evaluate our models. [5] Precisely, it calculates the likelihood of each choice in the question, and selects the answer with maximum likelihood, then 'accuracy' is used as the evaluation metric. The overall performance is calculated as the average accuracy of the model in 57 domains. Routoo models will be made publicly available for the AI community to test on various benchmarks.

**Baselines.** Our comparison includes a range of both open-source and closed-source LLMs. These comprise LLaMa2 (Touvron et al., 2023) models with 7b, 13b, and 70b parameters, Mistral 7b (Jiang et al., 2023a), Mixtral 8x7b (Jiang et al., 2024) (employing token-level MoE), alongside with GPT3.5 and GPT4 (OpenAI et al., 2024) as closed-source models. For additional comparison, we also included a k-nearest neighbors (KNN) method as a baseline. Further implementation details of this method are provided in Appendix B. [6]

**Architecture Setting.** For the pool of universe constructor ($\Omega$), we use top 1,000 available models of OpenLLM leaderboard (Beeching et al., 2023) [7], Then, we set the maximum number of serving models ($M$) to 56 in Equation 2, meaning that we use 56 models for the routing optimization, defined in Section 3.1. For the performance predictor, we use Mistral 7b (v0.1) (Jiang et al., 2023a) [8] as the query encoder. The number of levels in the evaluation score ($K$) is set to 2 for MMLU benchmark,

---

[5]Specifically, the following branch is used: `https://github.com/EleutherAI/lm-evaluation-harness/tree/b281b0921b636bc36ad05c0b0b0763bd6dd43463`.

[6]In relation to prior work, Lu et al. (2024), Ding et al. (2024), and Shnitzer et al. (2024) have publicly released a model and/or implementation for replication. However, akota et al. (2024) lacks a direct method for filtering a subset of LLMs from the complete set on the OpenLLM Leaderboard, making direct comparison infeasible.

[7]`https://huggingface.co/spaces/open-llm-leaderboard/open_llm_leaderboard`.

[8]`https://huggingface.co/mistralai/Mistral-7B-v0.1`.

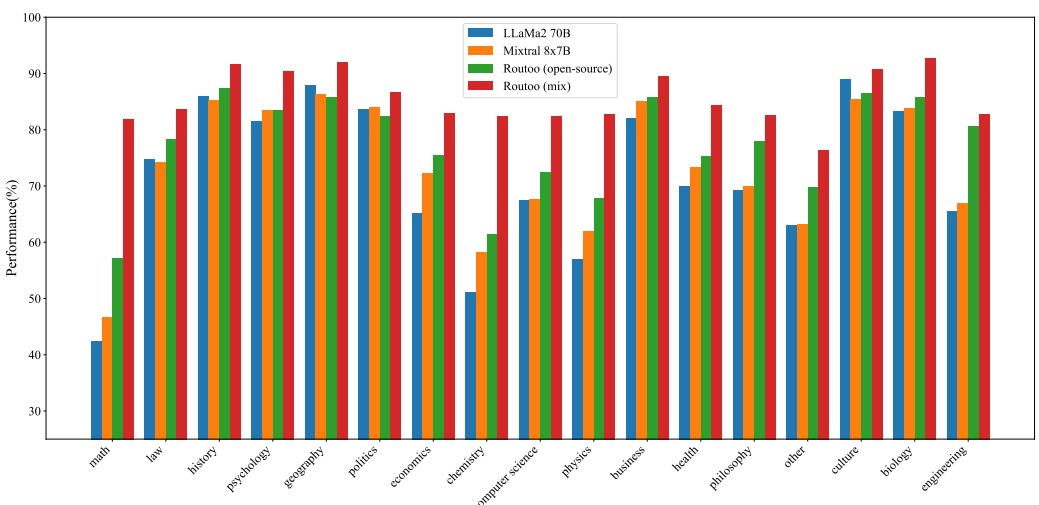

Figure 3: Per sub-category performances of our Routoo models and baselines on MMLU benchmark.

meaning that $s_{m_i,q_j}$ and $\hat{s}_{m_i,q_j}$ are either 0 or 1. [9] For fine-tuning, we apply LoRA method (Hu et al., 2021) with $r = 1024$, $\alpha = 16$, dropout $= 0.05$ on query, key, and value matrices.

## 4.2 Training Data Preparation

Since the MMLU dataset (Hendrycks et al., 2021) lacks a training set, we build synthetic data to train the models. The synthetic questions are originated by two approaches: Filtering available open-source datasets and generating synthetic queries using GPT4 model (OpenAI et al., 2024).

**Filtering Available Datasets.** We begin by collecting various multiple-choice QA datasets, such as ARC (Clark et al., 2018), MC-TEST (Richardson et al., 2013), OBQA (Mihaylov et al., 2018), RACE (Lai et al., 2017), and TruthfulQA (Lin et al., 2022). To identify questions with a high training signal (i.e., difficulty), we employed SOLAR-10.7B-v1.0 (Kim et al., 2024) [10] on this aggregated dataset, and calculate the evaluation score.[11] Queries on which the model under-performed (accuracy of 0) were retained, resulting in a set of 45,645 challenging training samples from an initial pool of 101,434. Additionally, 10,000 simpler questions were randomly selected from the initial dataset, bringing the total to 55,645 queries.

**Generation with GPT4.** To diversify our training dataset further, we generated 20,000 synthetic queries using GPT4 model. These queries are generated by using seed data, which randomly sampled from the dataset mentioned above. Detailed specifications of the input prompts used for generation are documented in Appendix C.

Final set of queries contains nearly 75,000 samples. Datatrove library[12] is used for decontamination. The set of queries in Universe Constructor ($Q : (q_1, q_2, \ldots, q_L)$) is created by randomly sampling 1,000 queries from the training dataset.

---

[9]As accuracy metric is used for MMLU benchmark in OpenLLM leaderboard (Beeching et al., 2023).

[10]Available in Huggingface platform: `https://huggingface.co/upstage/SOLAR-10.7B-v1.0`. We chose this model, as it is performing relatively well on MMLU section of OpenLLM benchmark (Beeching et al., 2023).

[11]MMLU evaluation system (introduced in Section 4.1) is used.

[12]`https://github.com/huggingface/datatrove`.

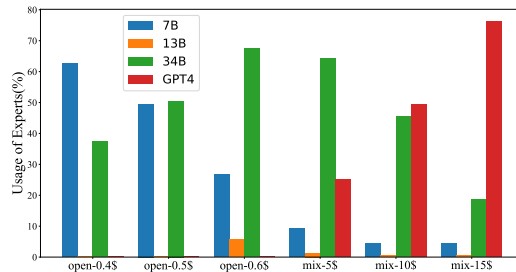

Figure 4: Routing distributions of different variants of Routoo models. The inference cost is provided for 1 million tokens.

## 4.3 MAIN RESULTS

**Our Routoo Models.** We present two variations of Routoo model: Routoo (open-source) and Routoo (mix). The former refers to a model that leverages 7b, 13b ,and 34b open-source models available on Huggingface as the input to our universe constructor. By additionally integrating GPT4 (OpenAI et al., 2024) to underlying models of Routoo (open-source), we create Routoo (mix). KNN variants (open-source and mix) are similarly defined.

**Overall Results.** Comparison of our models and baselines are illustrated in Table 1.[13] The cost of Routoo (open-source) is adjusted to match Mixtral 8x7b (Jiang et al., 2023a), the best generic open-source LLM (at the time of writing this paper). Our model significantly outperforms Mixtral 8x7b by 5.27% absolute points with the same inference cost. Also, our underlying models can be executed on 1 GPU (e.g. A100 with 40 GB memory), while Mixtral model requires access to high-end computing resources (e.g. GPUs with 100 GB memory). Compared to LLaMa2 70b, our model achieves significantly better performance (+6%), while reducing the cost by 33%. Impressively, Routoo (open-source) achieves competitive performance with GPT3.5 while reducing the cost by 73.3%. Compared to closed-source LLMs, our Routoo (mix) model significantly outperforms GPT3.5 by 14.9% absolute point. Routoo (mix) reaches competitive performance with GPT4, while reducing the cost by almost 50%. Additionally, Routoo models significantly outperforms KNN baselines, demonstrating that a LoRA fine-tuned LLM as performance predictor provides better estimations of the performance scores.

**Cost-Aware Selection.** Given that the cost-aware selector module can compute different routing distributions within a specified budget, we illustrate the curve of performance-cost in Figure 2 for both open-source and closed-source baselines, alongside with variations of Routoo models.[14] Notably, our Routoo (open-source) reaches the performance of Mixtral 8x7b (Jiang et al., 2024) while decreasing the cost by 33%. Interestingly, Routoo (mix) achieves better performance than GPT4 (86.64 vs. 86.40) while reducing the inference cost by nearly 25%.
In summary, our cost-aware selector effectively offers an optimal trade-off between cost and performance, enabling both open-source and mix variants of the Routoo model to outperform individual LLMs in terms of performance (or cost) during inference.

**Per Domain Comparison.** To further investigate the source of improvement in our models, we illustrates the distributions of performances on 17 sub-categories (Hendrycks et al., 2021) in Figure 3. A standout area of success is in STEM domains, such as mathematics and computer science, where Routoo particularly excels. This impressive performance is largely attributed to the incorporation of specialized small models (around 7b parameters) that are fine-tuned for tasks in mathematics (Yu et al., 2024; Shao et al., 2024) and coding (Rozière et al., 2024; Guo et al., 2024) by the community.

---

[13]Inference costs are calculated based on price documentation of the following providers at the time of writing the paper: https://www.together.ai, https://openai.com. For GPT3.5 and GPT4 costs, the average of input and output costs are considered.

[14]$\alpha$ parameter, defined in Section 3.3, is set to 1, 0.1, and 0.01 for data points from left to right of each sub-figure, respectively.

Furthermore, this approach facilitates the identification of domains where there is a scarcity of experts. Then, future research can focus on improving the performance of these areas by developing domain-specific effective experts.

**Routing Distribution.**   Figure 4 presents the aggregated percentage usage of expert models by size for each Routoo pricing tier ($ per 1M tokens). It reveals that higher-priced options tend to utilize larger models more frequently. A substantial inclusion of effective smaller, 7-billion parameter expert models significantly enhances the cost-to-performance efficiency. This suggests that strategically increasing the use of such smaller experts could offer a more economical solution while maintaining high-quality outputs.[15]

Finally, the cost of training a router is significantly lower than building foundational models e.g. GPT4 (OpenAI et al., 2024) and Mixtral (Jiang et al., 2024). This could pave a new path for building new frontiers at a much lower cost by integrating knowledge of current LLMs.

## 5   CONCLUSION AND FUTURE WORK

In this paper, we proposed our Routoo architecture, a lightweight LLM-based model that is designed to inteligently routes the input query to the most suitable expert model given other constraints e.g. cost. We evaluated our architecture on MMLU benchmark (Hendrycks et al., 2021), which is a MCQA dataset with 57 different domains coverage. Routoo (open-source) achieved competitive performance with Mixtral 8x7b model (Jiang et al., 2024) with two-thirds of the inference cost. Increasing the budget limitation allows Routoo (open-source) to outperform Mixtral model by 5% with the same level of cost. By integrating GPT4 (OpenAI et al., 2024) model to underlying LLMs, our Routoo (mix) achieved competitive accuracy with GPT4 while reducing the inference cost by 50%, and even surpassing it while saving 25% of the cost. In general, our Routoo models provide an efficient trade-off between cost and performance during the inference.

In future, the Routoo's ability to assess and understand the performance of existing models allows researchers to identify gaps in the AI landscape. It pinpoints domains where no existing expert excels or where larger models are inefficiently juggling tasks. This insight is invaluable, enabling us to strategically develop domain-specific models where they are needed most. Furthermore, our architecture facilitates easy integration of additional optimization criteria, including cost, speed, and privacy considerations.

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

## APPENDIX A    OPTIMAL ROUTING OF OPEN-SOURCE LLMS ON MMLU

In this experiment, we utilize all publicly available LLMs from the OpenLLM benchmark (Beeching et al., 2023) that have fewer than 34 billion parameters. Given that each query is directed to the most effective and cost-efficient model by an ideal performance predictor module within Routoo, the upper bound for the Routoo model in this scenario is nearly 97.5%. The following table shows the distribution of usage among LLMs of varying sizes:

| Model size | Usage (%) |
|---|---|
| 7b | 66.4 |
| 13b | 16.1 |
| 34b | 17.5 |

Table 2: The Distribution of usage among underlying LLMs with different sizes on MMLU (Hendrycks et al., 2021) test set.

The average size of the distribution above could be considered as an abstract model with approximately 13 billion parameters.

## APPENDIX B    KNN BASELINE IMPLEMENTATION

The algorithm of our k-nearest neighbors (KNN) method comprises the following steps:

- We utilized the "text-embedding-ada-002" model from OpenAI [16] to embed the question-answer pairs from the training data defined in Section 4.2. Each question-answer pair was converted into a string using the format: `<question>` [QUESTION] `</question>` `<answer>` [ANSWER] `</answer>`. For efficient inference implementation, we utilized the Faiss library (Douze et al., 2024; Johnson et al., 2019).

- During testing, the question field was converted into a string using the format: `<question>` [QUESTION] `</question>`.

- We then retrieved the top-5 related LLMs by applying cosine similarity function, then applied the cost-aware selection method defined in Section 3.3. Specifically, the estimated scores are the cosine similarity scores between the test question and the related question-answer pairs from the training data.

## APPENDIX C    SYNTHETIC DATA GENERATION BY GPT4

The following input prompt is used for generating synthetic data with GPT4 model (OpenAI et al., 2024):

```
Prompt for Generating Synthetic Queries

Generate {N} hard multiple-choice questions about {SUBJECT} field
    as defined in the following samples: \n\n
##Question:\n {} \n ##Choices:\nA. {}\nB. {}\nC. {}\nD. {}\n ##
    Answer: {}\n\n
##Question:\n {} \n ##Choices:\nA. {}\nB. {}\nC. {}\nD. {}\n ##
    Answer: {}\n\n
##Question:\n {} \n ##Choices:\nA. {}\nB. {}\nC. {}\nD. {}\n ##
    Answer: {}\n\n
##Question:\n {} \n ##Choices:\nA. {}\nB. {}\nC. {}\nD. {}\n ##
    Answer: {}\n\n
##Question:\n {} \n ##Choices:\nA. {}\nB. {}\nC. {}\nD. {}\n ##
    Answer: {}\n\n
```

---

[16]https://platform.openai.com/docs/guides/embeddings

For generation, we used OpenAI API [17] with temperature $= 0.7$. At each iteration, $N = 100$, and we run it for 200 times. SUBJECT is used when the seed samples contain subject field e.g. development set of MMLU benchmark (Hendrycks et al., 2021). Seed samples are chosen randomly from datasets introduced in Section 4.2, alongside with development set of MMLU benchmark. The synthetic data generation with GPT-4 incurred a cost of approximately $400 through the OpenAI API.

## APPENDIX D    DISTRIBUTION OF MODEL ASSIGNMENTS FOR MMLU

Further analysis of assignment distributions across 56 selected underlying models illustrated an interesting result. Almost all test queries are assigned to just six open-source LLMs (additional to GPT4). Following is the id list of Huggingface [18] models:

- `openaccess-ai-collective/mistral-7b-slimorcaboros`
- `kyujinpy/PlatYi-34B-Llama-Q`
- `berkeley-nest/Starling-LM-7B-alpha`
- `upstage/SOLAR-10.7B-Instruct-v1.0`
- `rishiraj/smol-7b`
- `upstage/SOLAR-10.7B-v1.0`

---

[17]`https://platform.openai.com/docs/overview`.
[18]`https://huggingface.co`

