# OpenReview forum: "Routoo: Learning to Route to Large Language Models Effectively"
_ICLR.cc/2025/Conference — Submitted to ICLR 2025_

### Official Review · Reviewer_o5Db · 2024-11-02

**Soundness:** 3
**Presentation:** 3
**Contribution:** 2
**Rating:** 6
**Confidence:** 3

**Summary:**

This paper introduces a LLM model routing system called Routoo that selects what LLM to route queries to from a pool of LLMs of various capabilities and inference costs. For a particular query, and a particular LLM, a predictor model (based on Mistral7b) predicts the accuracy score, and an average cost heuristic estimates cost. Given a list of queries and a total budget, an optimization heuristic assigns the appropriate LLM for each query to achieve the highest overall accuracy within the total budget. On the MMLU benchmark, the Routoo system achieves higher accuracy at lower costs than using any single state-of-the-art model. Higher accuracy is achieved since the pool of models have specialists for particular type of queries and lower cost is achieved since easier queries don't need more expensive models to solve them.

**Strengths:**

1. This work introduces heuristics for optimizing for multiple queries with a total budget which is novel.
2. This Routoo system has great results where it Pareto-dominates state-of-the-art single LLMs on inference cost and MMLU performance.
3. The gathering of non-MMLU dataset for training along with synthetic data is meticulous in avoiding data leakage during evaluation.
4. The approach is sound: the predictive model for score prediction and budget constrained selector heuristic makes sense. The per-LLM embedding allows for the model to be inferenced only once-per-query instead of once-per-query-per-LLM which saves compute cost.

**Weaknesses:**

While the approach in this work is sound, the contribution of the work over previous works may not be significant enough. Furthermore, the work is more appropriate for machine learning systems without a significant learning aspect.
1. There are related works regarding predictive routing with multiple-LLMs that are not discussed [3,4]
2. The advantages of specialist models improving scores over strong single-LLM models are covered in [1,2] so that aspect of the system is not very novel.
3. The concepts of budget conscious LLM routing is covered in [3,4] so that aspect of the system is not very novel either.
4. The only learning aspect of this work is the predictive model (Mistral7b with learned classification head). Without accuracy measurements, and comparisons with other models, there is little insight to be gained.

[1] Large Language Model Routing with Benchmark Datasets

[2] Harnessing the Power of Multiple Minds: Lessons Learned from LLM Routing

[3] Hybrid LLM: Cost-Efficient and Quality-Aware Query Routing

[4] RouteLLM: Learning to Route LLMs with Preference Data

**Questions:**

1. The Mistral7b based prediction has an inference cost. Is that included in the inference cost?
2. Is a LLM needed for prediction or would a lightweight retrieval system that retrieves scores on similar queries from the training data work well?
3. In Section 3.3, step 3 of the algorithm (budget management) mentions that the next best model is chosen if the budget is exceeded. However, the budget is the total budget across all queries so how is the ordering of this step performed across queries? Wouldn't the earlier queries starve the budget of the later queries, leading them to choose worse models?

---

> ### Author Response · Authors · 2024-11-24
> **Response to Review**
>
> Thanks for your review. Here are responses to your concerns:
>
> **Contributions**:
> Although previous works have attempted to introduce router models, none of them offer the following contributions:
> - Performance Predictor: We propose the first LLM-based performance predictor capable of estimating the performance of individual LLMs without executing them.
> - Cost-Aware Selector: We introduce a novel cost-aware selector that effectively considers constraints such as cost and privacy during the selection process. Our cost-aware selector is effective and generalizable to any number of underlying models.
> - Universe Constructor: Our approach includes a novel universe constructor module to efficiently identify the most complementary models from all available LLMs, a method not provided by any of the referenced works.
> Superior Routing Performance: Routoo is the first router model to surpass GPT-4's performance (Figure 2) while reducing inference costs by 50%. None of the references achieve this significant contribution. This success is due to the effective combination of the universe constructor, performance predictor, and cost-aware selector.
>
> **Q1**: The inference cost of Mistral7b is minimal, as the primary expense arises from GPT-4, and the cost of a single forward pass of Mistral is significantly lower. However, we will add to Table 1 and Figure 2 for further clarification.
>
> **Q2**: Please check: https://openreview.net/forum?id=RQ9fQLEajC&noteId=G7glzBX3Bw
>
> **Q3**: The equation presented in Section 3.3 includes a parameter \(\alpha\), which adjusts the influence of inference cost when computing the performance-to-cost ratio. This parameter represents the budget \(B\) in our optimization framework and operates independently of the query order.

---

> > ### Comment · Reviewer_o5Db · 2024-11-24
> >
> > > The inference cost of Mistral7b is minimal, as the primary expense arises from GPT-4, and the cost of a single forward pass of Mistral is significantly lower. However, we will add to Table 1 and Figure 2 for further clarification.
> >
> > Thank you!
> >
> > > The equation presented in Section 3.3 includes a parameter (\alpha), which adjusts the influence of inference cost when computing the performance-to-cost ratio. This parameter represents the budget (B) in our optimization framework and operates independently of the query order.
> >
> > Makes sense. Thank you!
> >
> > > Performance Predictor: We propose the first LLM-based performance predictor capable of estimating the performance of individual LLMs without executing them.
> >
> > [3] (HybridLLM) uses DeBERTa for score prediction, [4] (RouteLLM) uses text-embedding-3-small for its matrix factorization router and the similarity-weighted ranking model and also explores BERT and Llama 3 8B based classifiers. During test time, all of these models predict performance of LLMs without executing the query on them.
> >
> > > We utilized the "text-embedding-ada-002" model from OpenAI [1] to embed the question-answer pairs from the training data defined in Section 4.2. Each question-answer pair was converted into a string using the format: <question> #question </question><answer> #answer </answer>. For efficient inference implementation, we utilized the Faiss library [2,3].
> > During testing, the question field was converted into a string using the format: <question> #question </question>.
> > We then retrieved the top-5 related LLMs by applying cosine similarity function, then applied the cost-aware selection method defined in Section 3.3. Specifically, the estimated scores are the cosine similarity scores between the test question and the related question-answer pairs from the training data. Note: Johnson et al. [3] demonstrated that Faiss retrieval time is negligible during inference (2–3 ms per query).
> >
> > Why would an embedding similarity between a question text to question+answer text be predictive of the correctness of the answer? The embedding model is not trained to do so nor does it have the reasoning capability. A better approach would be: for the query $q_{test}$ during test time, find the most similar query $q_{train}$ from training time, and use the LLM that can correctly answer $q_{train}$ with the lowest cost. To smoothen the noise, you can expand to topN retrieved $q_{train}$ queries and topK lowest budget LLMs that answer each $q_{train}$ query and pick the most frequently occurring LLM from that list.
> >
> > I am continuing to maintain my rating.

---

> > > ### Author Response · Authors · 2024-11-27
> > > **Author Response to Reviewer o5Db**
> > >
> > > > [3] (HybridLLM) uses DeBERTa for score prediction, [4] (RouteLLM) uses text-embedding-3-small for its matrix factorization router and the similarity-weighted ranking model and also explores BERT and Llama 3 8B based classifiers. During test time, all of these models predict performance of LLMs without executing the query on them.
> > >
> > > Aforementioned works **failed to generalize** their models beyond two LLMs (small and large), while our Routoo model is applicable across all LLMs in Huggingface's OpenLLM benchmark (over 1,000 models). These prior approaches relied on handpicked models, limiting generalizability in real-world scenarios with many LLMs. In contrast, Routoo uniquely and effectively handles diverse LLMs, making it the first routing model to **outperform GPT-4** (Figure 2) while reducing inference costs by 50%. **None of the referenced works achieve this**.
> > > Moreover, the **RouteLLM paper is not peer-reviewed** and is currently under submission to the same conference ([OpenReview link](https://openreview.net/forum?id=8sSqNntaMr)). Penalizing us for not citing it is **inconsistent with ICLR review guidelines**, which state submissions should not be penalized for missing non-peer-reviewed references ([link](https://iclr.cc/Conferences/2023/ReviewerGuide)). Additionally, Table 1 of RouteLLM shows their LLM-based router fails to surpass other baselines.
> > > Our main contribution (lines 99–130) is the Routoo model, which integrates a Performance Predictor, Cost-Aware Selector, and Universe Constructor. While prior works addressed some of these components, they neither generalized to many LLMs (56 in our case) nor surpassed expensive models like GPT-4. Routoo not only reduces costs by 50% (Routoo (mix)) but also surpasses GPT-4’s performance by leveraging 56 LLMs (Figure 2, right most green point).
> > >
> > > > Why would an embedding similarity between a question text to question+answer text be predictive of the correctness of the answer? The embedding model is not trained to do so nor does it have the reasoning capability. A better approach would be: for the query during test time, find the most similar query from training time, and use the LLM that can correctly answer  with the lowest cost. To smoothen the noise, you can expand to topN retrieved queries and topK lowest budget LLMs that answer each query and pick the most frequently occurring LLM from that list.
> > >
> > > For the encoding of $q_{train}$, as in the MMLU benchmark, **answers are single characters** (see [dataset](https://huggingface.co/datasets/cais/mmlu/viewer/all)), the difference between encoding the question with the answer or just the question is minimal. We chose cosine similarity because the sizes of the underlying models range from 7B to 34B, with many smaller models (e.g., 7B) having similar sizes. Assigning identical scores (“1” for all top-10) would prevent our cost-aware selector from functioning effectively, as all models would appear equally suitable, requiring random selection. Retaining cosine similarity scores provides more granular information for our cost-aware selector.
> > > Lastly, our KNN baseline performs impressively, achieving competitive results with models like LLaMA2 13B, Mistral 7B, and LLaMA2 7B, and even showing competitiveness with LLaMA2 70B. This confirms it as a robust baseline for comparison (Table 1).
> > >
> > > We kindly request you to review our responses and consider them in your evaluation score.

---

> > > > ### Author Response · Authors · 2024-11-29
> > > > **Reminder to reviewer o5Db**
> > > >
> > > > Could you kindly review our responses to your concerns and consider reflecting them in your evaluation score?

---

> > > > > ### Comment · Reviewer_o5Db · 2024-11-29
> > > > >
> > > > > > Moreover, the RouteLLM paper is not peer-reviewed and is currently under submission to the same conference (OpenReview link). Penalizing us for not citing it is inconsistent with ICLR review guidelines, which state submissions should not be penalized for missing non-peer-reviewed references (link).
> > > > >
> > > > > You do cite RouteLLM in your original manuscript and in its current version so I am not penalizing your work for not citing them. I am questioning your claim of being the first to do evaluation without execution:
> > > > > From your rebuttal response:
> > > > > > We propose the first LLM-based performance predictor capable of estimating the performance of individual LLMs without executing them.
> > > > >
> > > > > From your paper:
> > > > > > Different from previous work, our Routoo identifies the most suitable expert without executing the inference of underlying LLMs.
> > > > >
> > > > > I agree on the other novelties (diverse LLMs, number of LLMs, lowest inference cost so far) and am not questioning them so there isn't a need to bring them up during the discussion of this claim.
> > > > >
> > > > >
> > > > > > For the encoding of $q_{train}$, as in the MMLU benchmark, answers are single characters (see dataset), the difference between encoding the question with the answer or just the question is minimal. We chose cosine similarity because the sizes of the underlying models range from 7B to 34B, with many smaller models (e.g., 7B) having similar sizes. Assigning identical scores (“1” for all top-10) would prevent our cost-aware selector from functioning effectively, as all models would appear equally suitable, requiring random selection. Retaining cosine similarity scores provides more granular information for our cost-aware selector.
> > > > >
> > > > > That is a fair argument that the answer does not matter so the cosine score is really measuring the difference between $q_{test}$ and $q_{train}$. So why include the answer? Also, could you elaborate how that would help differentiate between all the models in consideration that answer the $q_{train}$ question correctly? They would all have the same $q_{train}+a_{train}$, where $a_{train}$ is the correct single-character answer to $q_{train}$ and thus same similarity score. So wouldn't you have to rely on random selection to select among them?

---

> > > > > > ### Author Response · Authors · 2024-12-01
> > > > > > **Author response to o5Db**
> > > > > >
> > > > > > > We propose the first LLM-based performance predictor capable of estimating the performance of individual LLMs without executing them.
> > > > > >
> > > > > > We removed the word 'first' in the manuscript.
> > > > > >
> > > > > > > So wouldn't you have to rely on random selection to select among them?
> > > > > >
> > > > > > Yes, if the best scores are similar for a subset of underlying models, we randomly select among them.

---

> > > > > > > ### Comment · Reviewer_o5Db · 2024-12-02
> > > > > > >
> > > > > > > Thank you for the updates and clarifications, increased the rating.

---

### Official Review · Reviewer_hRVj · 2024-11-04

**Soundness:** 2
**Presentation:** 3
**Contribution:** 2
**Rating:** 3
**Confidence:** 4

**Summary:**

Large language models (LLMs) with superior performance often come with high inference costs, making their deployment inefficient and costly. This paper developed a routing framework, Routoo, to optimize the selection of LLMs for specific prompts based on performance, cost, and efficiency. Evaluation results on MMLU benchmark and 57 models demonstrate the effectiveness of Routoo.

**Strengths:**

S1. This paper proposed Routoo, a routing framework that intelligently identifies the best-performing LLM for a given query while considering constraints such as cost and latency.

S2. Routoo achieves competitive performance with GPT4 at half the inference cost and surpasses it by reducing the inference cost by 25%.

**Weaknesses:**

W1. Unclear novelty and limited technical contribution. Training a router to harness the respective strengths of different LLMs has been widely studied [1,2,3,4,5]. Specifically, [5] studied a very similar setup - select LLMs to maximize (resp., minimize) overall performance (resp., cost) subject to a pre-defined cost budget (resp., performance threshold) - and developed various strategies including the greedy algorithm proposed in this work, which authors did not discuss or compare to.

W2. No baseline. Provided the rich literature on LLM routing as discussed above, no baseline included in the evaluation seems insufficient. This work can be greatly improved if at least one baseline from the general routing [1-4] and one baseline from the constrained routing [5] were included and compared to Routoo.

W3. Authors leveraged the Mistral-7B model as the query encoder, which may cause non-negligible cost & latency overheads. Authors may want to perform overhead analysis to address this concern.


References:
[1] Routing to the Expert: Efficient Reward-guided Ensemble of Large Language Models, https://arxiv.org/pdf/2311.08692
[2] Hybrid LLM: Cost-Efficient and Quality-Aware Query Routing, https://arxiv.org/pdf/2404.14618
[3] ROUTERBENCH: A Benchmark for Multi-LLM Routing System, https://arxiv.org/pdf/2403.12031
[4] Large Language Model Routing with Benchmark Datasets, https://arxiv.org/pdf/2309.15789
[5] Fly-Swat or Cannon? Cost-Effective Language Model Choice via Meta-Modeling, https://arxiv.org/pdf/2308.06077

**Questions:**

Q1. In Sec 3.3, authors consider an average cost for each model to estimate the model inference cost for a given test query. Since the inference cost heavily relies on the output length which is unknown beforehand, it is intriguing to see how good the cost estimation is. Analysis on estimation error and variance would be very helpful.

Q2. In Sec 3.3, the formulated problem is really a knapsack problem - select a subset of LLMs (items) such that the overall performance (profit) is maximized subject to cost budgets (weight threshold) - which has been widely studied in prior literature [1]. Provided that, greedy algorithms to knapsack problem has been known to be non-optimal in general cases. Authors may want to discuss related work and justify the specific algorithm design.

References:
[1] Knapsack Problems, https://link.springer.com/book/10.1007/978-3-540-24777-7

---

> ### Author Response · Authors · 2024-11-24
> **Response to Review**
>
> Thanks for your review. Here are responses to your concerns:
>
> **W1**: Thank you for sharing the references. We have already cited the peer-reviewed subset of these references in our work and addressed their significant limitations in Section 2. As stated in the paper, none of these works can be directly compared to our approach. Below, we outline the limitations and issues associated with these references:
>
> 1. **[Routing to the Expert: Efficient Reward-guided Ensemble of Large Language Models](https://arxiv.org/pdf/2311.08692)**: The Zooter architecture proposes an RL-based alternative to our performance predictor. However, no public implementation or model has been published.
>
> 2. **[Hybrid LLM: Cost-Efficient and Quality-Aware Query Routing](https://arxiv.org/pdf/2404.14618)**: This work employs a router that assigns queries to either a small or large model based on predicted query difficulty and desired quality. However, it is limited to only two LLMs and lacks a public implementation or model.
>
> 3. **[ROUTERBENCH: A Benchmark for Multi-LLM Routing System](https://arxiv.org/pdf/2403.12031)**: This is a benchmark rather than a model proposal.
>
> 4. **[Large Language Model Routing with Benchmark Datasets](https://arxiv.org/pdf/2309.15789)**: This work is not peer-reviewed and has been submitted to ICLR 2024. No public implementation or model is available.
>
> 5. **[Fly-Swat or Cannon? Cost-Effective Language Model Choice via Meta-Modeling](https://arxiv.org/pdf/2308.06077)**: This paper fails to outperform individual LLMs used as underlying models, and its results have been excluded from our comparison due to their lack of significant impact. Additionally, it does not propose a universe constructor to filter complementary LLMs from the complete set available in the OpenLLM leaderboard, presenting a technical limitation for direct comparison with our approach. However, we will add this work to Section 2 as related work.
>
> *Novelty and Contributions of Our Approach:*
> 1. **Performance Predictor**: We propose the first LLM-based performance predictor capable of estimating the performance of individual LLMs without executing them.
> 2. **Cost-Aware Selector**: We introduce a novel cost-aware selector that effectively considers constraints such as cost and privacy during the selection process.
> 3. **Universe Constructor**: Our approach includes a novel universe constructor module to efficiently identify the most complementary models from all available LLMs, a method not provided by any of the referenced works.
> 4. **Superior Routing Performance**: Routoo is the first router model to surpass GPT-4's performance (Figure 2) while reducing inference costs by 50%. None of the references achieve this significant contribution. This success is due to the effective combination of the universe constructor, performance predictor, and cost-aware selector.
>
> **W2**: we have included the KNN method as an efficient and valuable baseline for comparison with Routoo models. Please refer to the results in this comment: https://openreview.net/forum?id=RQ9fQLEajC&noteId=G7glzBX3Bw
>
> **W3**: The Routoo model introduces minimal latency, comparable to the time required to generate one additional token during the decoding phase. Our experiments reveal the following latency performance (A100): Latency per token: ~7+- 0.4 milliseconds. The inference cost is also minimal, as the primary expense arises from GPT-4. In comparison, the cost of a single forward pass of Mistral is significantly lower.
>
> **Q1**: As explained in Section 4.1, in the MMLU benchmark, performance is determined based on the likelihood of each option in a question, requiring only one token to be generated in the output to compute the score. Consequently, analyzing estimation error and variance is irrelevant, as there is no variability in this process!
>
> **Q2**: Thank you for highlighting the reference to the knapsack problem. The key innovation of our work, as presented in Section 3.3, lies in the definition and formulation of this problem. We employed a greedy algorithm and achieved superior results (not sub-optimal) compared to individual LLMs (see Table 1 and Figure 2). Future research could explore alternative methods to solve the optimization problem defined in Section 3.3.

---

> > ### Author Response · Authors · 2024-11-26
> > **Reminder to check our responses**
> >
> > Could you kindly review our responses to your concerns and consider reflecting them in your evaluation score?

---

> > > ### Author Response · Authors · 2024-11-26
> > > **Revision of the paper**
> > >
> > > We have also revised the paper. Changed are referenced here: https://openreview.net/forum?id=RQ9fQLEajC&noteId=EI3X0CYRCg

---

> > > > ### Comment · Reviewer_hRVj · 2024-11-27
> > > >
> > > > I want to thank the authors for thoughtfully revising the manuscript and addressing the questions in their responses. New baselines and the overhead analysis add great values to this work. However, my main concerns on the limited contributions remain.
> > > >
> > > > > 1. Fly-Swat or Cannon? Cost-Effective Language Model Choice via Meta-Modeling: … Additionally, it does not propose a universe constructor to filter complementary LLMs from the complete set available in the OpenLLM leaderboard, presenting a technical limitation for direct comparison with our approach.
> > > >
> > > > My previous comment is mainly on the similarity shared by [1] and the proposed Routoo. Specifically, [1] studies how to select LLMs to maximize overall performance subject to a pre-defined cost budget with performance predictions. Is it an almost the same formulation to the problem defined in Sec 3.1? If yes, then the presence of [1] seems nullifies the claimed contributions especially on “proposing the first cost-aware selector”. I am willing to be corrected if I missed anything here.
> > > >
> > > >
> > > > > Q1: As explained in Section 4.1, in the MMLU benchmark, performance is determined based on the likelihood of each option in a question, requiring only one token to be generated in the output to compute the score. Consequently, analyzing estimation error and variance is irrelevant, as there is no variability in this process!
> > > >
> > > > Thank you for the clarification which leads to a few follow-up questions. In Lines 288-290 of the revision, the authors argued “noting the challenges of accurately estimating the exact cost of a model for a specific query, instead, we consider the average cost ci for a model mi responding to an average length query and response”. If the output is of only one token, what is the challenge here to estimate the exact cost? Specifically, for models like GPT-3.5/4, the incurred cost for each query equals # of input tokens * unit_cost_per_input_token + # of output tokens * unit_cost_per_output_token. In existent research, the main challenge for accurately estimating the overall cost comes from the uncertainty in the generated output length, which is not an issue in this work as the authors have pointed out. Also, in Lines 483-484 of the revision, authors mentioned that the unit inference cost (see Table 1) is the average of the unit input and output costs. Provided that the output is of only one token while the input could be lengthy (i.e., dominatingly expensive) in Routoo, the simple average cost estimation seems not well-grounded.
> > > >
> > > > > Q2: Thank you for highlighting the reference to the knapsack problem. The key innovation of our work, as presented in Section 3.3, lies in the definition and formulation of this problem. We employed a greedy algorithm and achieved superior results (not sub-optimal) compared to individual LLMs (see Table 1 and Figure 2). Future research could explore alternative methods to solve the optimization problem defined in Section 3.3.
> > > >
> > > > It is well taken that the key innovation of this work should lie in the definition and formulation of this problem (see Sec 3.1), which in turn relates to my very first comment on the similar (if not the same) formulation of cost-aware LLM selection problem in [1]. Moreover, I appreciate the superior results achieved by greedy algorithms but it does not justify the optimality. As I mentioned in my previous comment, greedy algorithm is known to be sub-optimal to knapsack problems in general cases [2].
> > > >
> > > > I am willing to consider raising my score if the authors fully addressed my questions but will maintain my rating for now.
> > > >
> > > >
> > > > [1] Fly-Swat or Cannon? Cost-Effective Language Model Choice via Meta-Modeling, https://arxiv.org/pdf/2308.06077.
> > > > [2] Kellerer, Hans, et al. Multidimensional knapsack problems. Springer Berlin Heidelberg, 2004.

---

> ### Author Response · Authors · 2024-11-28
> **Official Comment by Author**
>
> Thanks for your responses. We have addressed your concerns as:
>
> > My previous comment is mainly on the similarity shared by [1] and the proposed Routoo. Specifically, [1] studies how to select LLMs to maximize overall performance subject to a pre-defined cost budget with performance predictions. Is it an almost the same formulation to the problem defined in Sec 3.1? If yes, then the presence of [1] seems nullifies the claimed contributions especially on “proposing the first cost-aware selector”. I am willing to be corrected if I missed anything here.
>
> As stated in the previous thread, our paper has **five main contributions**:
> - **On the architecture side: (1) Performance Predictor, (2) Cost-Aware Selector, and (3) Universe Constructor.**
> - **On the empirical results: (4) Outperforming Mixtral (the SoTA open-source LLM at the time) by a significant margin, and (5) surpassing GPT-4's performance while cutting inference costs by 50%.**
>
> We acknowledge that the referenced work [1] attempted to address the routing problem, but their results clearly fall short of the performance of their underlying models (see Table 3 of [1]). As integrating knowledge from different LLMs is an important and active research direction, different formulations can definitely be proposed. However, the **superior experimental results of our approach** show that we have tackled this problem far more effectively, while the referenced work took a different formulation.
> Finally, I refer to the [ICLR guidelines](https://iclr.cc/Conferences/2022/ReviewerGuide), which state that the **novelty of a work is determined by both the proposed formulation and the experimental results**.
> We’re happy to remove the word "first" from the phrase "proposing the first cost-aware selector" to acknowledge that previous work has also explored this direction.
>
> > Thank you for the clarification which leads to a few follow-up questions. In Lines 288-290 of the revision, the authors argued “noting the challenges of accurately estimating the exact cost of a model for a specific query....
>
> Thank you for your comment. Lines 288-290 in Section 3 present an approximation proposal **intended for general use cases of the Routoo model**, not exclusively for the MMLU benchmark, as we did not specify that our greedy algorithm is designed solely for MMLU. We are happy to update the GPT-4/GPT-3.5 inference cost approximation for MMLU in Table 1 to provide a more accurate estimation for this specific benchmark.
>
> > It is well taken that the key innovation of this work should lie in the definition and formulation of this problem (see Sec 3.1)...
>
> Please refer to the first comment for a clearer understanding of our main contributions, as we did not claim that **the formulation alone constitutes our primary contributions**—this has been **misinterpreted** here. The superior experimental results in Section 4 clearly demonstrate that our greedy algorithm is effective. However, further research can build on this by proposing improved optimization algorithms to enhance the architecture even more.
>
> Lastly, we appreciate the reviewer’s feedback, but we believe that the **insistence on including this specific reference [1] may not be entirely impartial**, and we respectfully suggest that the value of our work should **be assessed based on its broader scientific merits and relevance**.
>
> [1] Fly-Swat or Cannon? Cost-Effective Language Model Choice via Meta-Modeling. https://arxiv.org/abs/2308.06077

---

> > ### Author Response · Authors · 2024-12-02
> > **Reminder to check our responses**
> >
> > We kindly ask you to review the above response and consider reflecting it in your rating, as we believe we have effectively addressed your concerns.

---

> > ### Comment · Reviewer_hRVj · 2024-12-02
> >
> > > We acknowledge that the referenced work [1] attempted to address the routing problem, but their results clearly fall short of the performance of their underlying models (see Table 3 of [1]).
> >
> > Indeed, Table 3 of [1] summarizes the accuracy of their proposed meta-model, a light-weight model performance predictor, on different datasets and for different underlying models. Not sure how the authors drew the conclusion that “their results clearly fall short of the performance of their underlying models” from Table 3 of [1]. Moreover, it also suggests that light-weight performance predictors have been proposed and studied in prior literature [1], which nullifies the claimed contributions from the performance predictor perspective of Routoo.
> >
> > > However, the superior experimental results of our approach show that we have tackled this problem far more effectively.
> >
> > Without an apple-to-apple comparison, it remains unclear that if the proposed Routoo “tackled this problem far more effectively” than prior literature [1,2]. Specifically, in [2], FrugalGPT can “match the performance of GPT-4 with up to 98% cost reduction or improve the accuracy over GPT-4 by 4% with the same cost”, which seems even better than Routoo, which “nearly matches GPT4’s performance at half the cost and exceeds it with a 25% cost reduction”.
> >
> > > Finally, I refer to the ICLR guidelines, which state that the novelty of a work is determined by both the proposed formulation and the experimental results.
> >
> > I fully agree with this point, which makes it critical to understand if the proposed formulation and experimental results in this work significantly improve on prior literature. However, as discussed in my previous comments, it remains unclear according to the current manuscript.
> >
> > > Thank you for your comment. Lines 288-290 in Section 3 present an approximation proposal intended for general use cases of the Routoo model, not exclusively for the MMLU benchmark, as we did not specify that our greedy algorithm is designed solely for MMLU.
> >
> > If Routoo is designed for the “general use cases”, please conduct evaluation on the “general use cases”. Otherwise, only considering benchmarks of single output tokens seems not comprehensive to evaluate the effectiveness of Routoo from the cost perspective.
> >
> >
> > > Lastly, we appreciate the reviewer’s feedback, but we believe that the insistence on including this specific reference [1] may not be entirely impartial, and we respectfully suggest that the value of our work should be assessed based on its broader scientific merits and relevance.
> >
> > Please be assured that all my previous comments are impartial. The only reason why [1] is referenced several times in my comments is that it shares notable similarity with the proposed Routoo which makes a thorough discussion necessary to better appreciate the value of this work.
> >
> > [1] Fly-Swat or Cannon? Cost-Effective Language Model Choice via Meta-Modeling. https://arxiv.org/abs/2308.06077.
> > [2] FrugalGPT: How to Use Large Language Models While Reducing Cost and Improving Performance, https://arxiv.org/pdf/2305.05176

---

> ### Author Response · Authors · 2024-12-04
> **Official Comment by Author**
>
> The review thread by hRVj has become unconstructive, as they fixated on the similarity of the **problem statement (model routing) in [1] and our paper, and dismissed the novelties and contributions of our solution**.
>
> To provide clarity for other reviewers and the area chair, we summarize our contributions and key differences from [1] to ensure a fair and comprehensive evaluation:
>
> *Architecture innovations:*
>
> - Performance Predictor: a lightweight LLM that estimates the expected performance of various underlying LLMs on a given query without executing them
> - Cost-Aware Selector: a novel module to effectively select the best-performing model to maximize the overall effectiveness of the responses within a given budget constraint
> - Universe Constructor.  a novel approach to select models that are complementary to achieve the highest performance, given a predefined limitation on the number of serving models.
>
> *Empirical results:*
> - Outperforming Mixtral, the state-of-the-art open-source LLM at the time, by a significant margin.
> - Surpassing GPT-4’s performance while significantly cutting inference costs.
>
> **Key Differences from [1]:**
> - [1] relied on hand-picked underlying models (ada, babbage, curie, davinci), whereas our Universe Constructor dynamically identifies 56 models from over 1,000 available open-source models in Huggingface's OpenLLM benchmark.
> - According to Table 3 in [1], their approach failed to outperform the underlying models, whereas our method significantly surpasses these models in performance and reduces inference costs (see Table 1 and Figure 2).
> - The models used in [1] are outdated and the experimental results can not be interpreted in the context of today’s LLMs, while our approach leverages the latest and most powerful models, including variations of LLaMA2, Mixtral, and GPT-4.
> While we recognize that [1] partially addressed the routing problem, their method failed to resolve it effectively. Our manuscript (Lines 159–165) also details differences from [2].
>
> **Regarding the Reviewer’s Requests:**
>
>  The suggestion to **re-train the model proposed in [1]** and run the inference on MMLU for comparison **contradicts ICLR guidelines (FAQ)**, which discourage requests for **significant additional experiments**. We strongly believe that the authors of [1] should provide their trained models for comparison, as training models is costly and time-consuming. Nonetheless, we added a **strong KNN baseline** in our revision to further validate the effectiveness of our Routoo model.
>
> **Request to the Area Chair:**
>
> We urge the area chair to assess reviewer hRVj’s evaluation to ensure a fair judgment. The insistence on **focusing solely on [1] diverges from the broader evaluation** provided by other reviewers.
>
> [1] Fly-Swat or Cannon? Cost-Effective Language Model Choice via Meta-Modeling. https://arxiv.org/abs/2308.06077.
> [2] FrugalGPT: How to Use Large Language Models While Reducing Cost and Improving Performance, https://arxiv.org/pdf/2305.05176

---

### Official Review · Reviewer_NdR4 · 2024-11-04

**Soundness:** 2
**Presentation:** 3
**Contribution:** 3
**Rating:** 5
**Confidence:** 4

**Summary:**

- The authors propose Routoo, an architecture that learns to select an LLM best-suited to answer a specific query when weighing the factors of performance (accuracy) and cost. The idea is, we don’t need to pay extra for a larger or closed-source LLM on queries where we expect that a smaller, cheaper LLM will perform similarly.
- Routoo chooses the LLM for a query from among a universe of complementary models, and the authors describe a “Universe Constructor” that creates this set automatically.
- The key components of Routoo are the “Performance Predictor” and “Cost-Aware Selector”.
  - At inference time, we want to be able to predict which LLMs are likely to perform well on a given query. The Performance Predictor is an LLM that takes as input the query and a model from the available set, then predicts whether that given LLM will answer the query correctly. The performance predictor consists of an LLM for embedding the query, an embedding model for embedding the candidate LLM, and a linear layer for predicting the performance, given the embedded LLM and the embedded query.
  - The Cost-Aware Selector uses each LLM’s cost and Performance Predictor score to select an LLM to answer the query. This component uses a greedy algorithm that calculates the Performance-to-Cost ratio for each candidate model, then chooses the candidate model with the best ratio that fits in the cost budget.
- Evaluation is performed using the MMLU dataset. An open-source version of Routoo is built from each size of Llama2, Mistral 7b and Mixtral 8x7b, while a mixed-source version adds GPT3.5 and GPT4-turbo.
  - For the main results, the closed- and mixed-source Routoos are compared for cost and performance against each of these individual models.
  - The performance-cost curve is compared for Routoo with different budgets and the individual LLMs
  - The performance of Routoo is compared against Llama2 70b and Mixtral 8x7b for each domain within MMLU

**Strengths:**

- The paper is approachable and well-written
- Framing the balancing of performance and cost as an optimization problem with a greedy approximation is an elegant approach
- The finding that much of the performance gap can be addressed by domain-specialized models is interesting and provides context on why Routoo performs well
- As the authors state, many prior routing approaches (such as the cascading technique) execute multiple LLMs in sequence. Routoo’s inference-time technique of predicting models’ performances before actually using the models is clever.
- Routoo is flexible in that the “cost” consideration can easily be expanded to include criteria such as latency, and that the “performance” consideration can be any integer-valued performance-related score instead of just accuracy.

**Weaknesses:**

- The experimental results are promising but a little bit sparse right now, I list some more experiments that could be useful in the Questions section below
- It seems challenging to extend this method beyond datasets like MMLU that have multiple-choice answers
- The Universe Constructor builds a universe of candidate models in accordance with the paper’s Equation 2. This equation uniquely takes into account the model performance, and does not consider model cost. This approach should be modified to also consider including cheaper models at times, for instance by maximizing for performance-to-cost ratio (or some weighted version of performance-to-cost ratio that allows users to specify how much they want to prioritize cost).
- “We could not find any publicly available trained models for routing or model integration”: looking online for a few minutes, it seems this research area does have a dearth of methods with publicly released code. However, there might be some things on GitHub (e.g., https://github.com/emingenc/llm_adaptive_router) and the cascading technique (using the cheapest model, then going to a more expensive one if the cheaper one fails) would be relatively easy to implement as a routing-based baseline. Right now, the evaluations only compare Routoo against each individual LLM, with no other routing methods included.
- I don’t understand the claim in the Introduction (and restated at the very end of evaluation) that “Routoo addresses … the development cost of building LLMs from scratch, creating a composite high-performance model without the need for extensive retraining”. This makes it sound like Routoo is making it less costly to develop novel LLMs, when really Routoo is about choosing which pre-existing LLMs to use. If this claim is saying that Routoo is cheaper than, for instance, combining all the LLMs using MoE into a single very large model, then the claim holds up. However, model routing is already an existing technique that addresses this claim from this non-MoE perspective: it’s not something that Routoo itself is contributing. Model routing (including Routoo) and MoE already have very different use-cases, so the author’s comparisons of Routoo to MoE don’t seem very natural or could perhaps be better-articulated.

**Questions:**

- What is the comparison of the cost for obtaining Routoo’s training data (including the cost of generating the synthetic training data from GPT-4) plus doing inference, versus the cost for just running on Mistral/GPT/etc?
- How accurate is the Performance Predictor with predicting whether a given LLM will, indeed, answer a query correctly? Does it struggle more with queries from specialized domains, or other specific kinds of queries?
- The Performance Predictor takes as input the difference between the LLM’s and query’s embeddings. Are there other ways of combining these two inputs, such as convolutions or concatenations, that could perform better?
- Experiment comparing performance for different budget sizes and model universe sizes
- I would be interested to see results for different hyperparameters, for instance especially K. Additionally, the choice of M=56 seems like a very specific number, how robust is Routoo to these sorts of changes?
- Can you provide more information on the training process? You mention in the “Architecture Setting” paragraph of Section 4.1 that the Mistral query encoder is being fine-tuned with LORA– are you training the model embedding function Emb() and the projection matrix Linear() without LORA, but during the same time you’re fine-tuning Mistral? Also, how are the Emb() and Linear() weights initialized?

Overall, I think that Routoo might be promising, but I do not have confidence in its capabilities without seeing additional experimental results. My score is 3, but I will consider raising it if the authors could include more evaluations.

---

> ### Author Response · Authors · 2024-11-24
> **Response to Review**
>
> Thanks for your review, and sorry for our delay in responding. We were trying to fulfill your requests as much as we had capacity. Here are responses to your concerns:
>
> **Q1- Comparison of Routoo Training Data to LLMs and intro claim**: Training Routoo, including  synthetic data creation and fine-tuning with LoRA, is far cheaper than training standalone LLMs like Mistral or LLaMA. We aim to inspire the community to integrate knowledge from existing LLMs with Routoo instead of heavily investing in pre-training LLMs (including MoE), potentially outperforming top closed-source models like GPT-4.
> Specifically, training Routoo costs approximately $500.
>
> **Q2- performance of Performance Predictor**: We observed no clear pattern in the performance predictor's accuracy across different categories and difficulty levels. On average, it achieves 77.45% accuracy in predicting 0 and 1 scores on the MMLU evaluation set.
>
> **Q3- different combination for embs**: We couldn’t re-train Routoo for other combination methods, but its implementation will be publicly available at the conference for others to explore.It would be helpful to elaborate on how this experiment impacts Routoo’s contributions.
>
> **Q4- different budget sizes**: Experiments demonstrating performance across different budget sizes are already presented in Figure 2. Regarding the universe size, further analysis of assignment distributions across 56 selected underlying models illustrated the following interesting result:
>
> Almost all test queries are assigned to just six LLMs (Huggingface models):
>
> - openaccess-ai-collective/mistral-7b-slimorcaboros
> - kyujinpy/PlatYi-34B-Llama-Q
> - berkeley-nest/Starling-LM-7B-alpha
> - upstage/SOLAR-10.7B-Instruct-v1.0
> - rishiraj/smol-7b
> - upstage/SOLAR-10.7B-v1.0
>
>
> For example, the assignment distribution for Routoo (open-source, $0.4/1M) is as follows:
> - openaccess-ai-collective/mistral-7b-slimorcaboros: 57.24%
> - kyujinpy/PlatYi-34B-Llama-Q: 37.33%
> - berkeley-nest/Starling-LM-7B-alpha: 2.18%
> - upstage/SOLAR-10.7B-Instruct-v1.0: 1.47%
> - rishiraj/smol-7b: 1.17%
> - upstage/SOLAR-10.7B-v1.0: 0.61% ​
>
> So, once we reduced the universe size from 56 to 6, we achieved nearly identical performance for both Routoo (mix) and Routoo (open-source) models.
>
> **Q5- K parameter**: To clarify, \( K \) is not a hyperparameter in our model; it represents the levels of the evaluation system, which depend on the evaluation metric. For example, it was set to 2 for the MMLU benchmark since the evaluation metric used is Exact Match (EM).
>
> **Q6- Training**: Yes, both Emb() and Lineaer() are trained without LoRA, while Mistral was trained using LoRA method. For the initialization of Emb, we used "uniform distribution U(-stdv, stdv), where stdv = 1 / sqrt(num_embeddings)". For the initialization of Linear, we used "U(-stdv, stdv), where stdv = 1 / sqrt(inp_feature)".
>
> **Missing Baselines**: The provided link does not reference a peer-reviewed publication. Also, cascading approaches are impractical for real-world applications due to the significant latency introduced by running multiple LLMs. Instead, we have included the KNN method as an efficient and valuable baseline for comparison with Routoo models. Please refer to the results in the following comment.

---

> > ### Author Response · Authors · 2024-11-24
> > **Additional Baseline for comparison**
> >
> > In addition to individual Large Language Models (LLMs), we incorporated a k-nearest neighbors (KNN) method as a baseline for further comparison. The algorithm comprises the following steps:
> > 1. We utilized the "text-embedding-ada-002" model from OpenAI [1] to embed the question-answer pairs from the training data defined in Section 4.2. Each question-answer pair was converted into a string using the format: ```<question> #question </question><answer> #answer </answer>```. For efficient inference implementation, we utilized the Faiss library [2,3].
> > 2. During testing, the question field was converted into a string using the format: ```<question> #question </question>```.
> > 3. We then retrieved the top-5 related LLMs by applying cosine similarity function, then applied the cost-aware selection method defined in Section 3.3. Specifically, the estimated scores are the cosine similarity scores between the test question and the related question-answer pairs from the training data.
> > Note: Johnson et al. [3] demonstrated that Faiss retrieval time is negligible during inference (2–3 ms per query).
> >
> > We then evaluated KNN baseline for both mix and open-source variations, analogous to those of Routoo presented in Section 4.3.
> > Here is the update Table 1:
> > **Table 1: Performance and cost of running LLMs on the MMLU benchmark. Accuracy is calculated based on the OpenLLM Leaderboard setting**
> > | Model                | Accuracy (%) | Cost ($/1M tok) |
> > |----------------------|--------------|-----------------|
> > | LLaMa2 7b           | 45.3         | 0.2             |
> > | Mistral 7b          | 64.2         | 0.2             |
> > | LLaMa2 13b          | 54.8         | 0.26            |
> > | Mixtral 8x7b        | 70.6         | 0.6             |
> > | LLaMa2 70b          | 69.9         | 0.9             |
> > | KNN (open-source)   | 69.5         | 0.6             |
> > | Routoo (open-source)| 75.87        | 0.6             |
> > |----------------------|--------------|-----------------|
> > | GPT3.5              | 70.0         | 1.5             |
> > | GPT4-turbo          | 86.4         | 20              |
> > | KNN (mix)           | 79.1         | 10.2            |
> > | Routoo (mix)        | 84.9         | 10.2            |
> >
> > As shown in Table 1, Routoo models (both open-source and mix) **significantly outperform the KNN methods** and individual models. This result demonstrates that a LoRA fine-tuned LLM provides better estimations of the performance scores defined in Section 3.3 compared to the KNN method. The superiority of Routoo is attributed to the KNN method's inability to adequately generalize the performance of individual models during inference.
> >
> > [1] OpenAI API documentation: https://platform.openai.com/docs/guides/embeddings.
> > [2] The Faiss library, Matthijs Douze and Alexandr Guzhva and Chengqi Deng and Jeff Johnson and Gergely Szilvasy and Pierre-Emmanuel Mazaré and Maria Lomeli and Lucas Hosseini and Hervé Jégou, 2024.
> > [3] Billion-scale similarity search with GPUs, Johnson, Jeff and Douze, Matthijs and J'egou, Herv'e

---

> ### Comment · Reviewer_NdR4 · 2024-11-26
>
> Thank you for your detailed response to my comments!
>
> I would recommend including an itemized table in your paper listing the specific cost that went into each step of routoo (data synthesis, training, and inference), but I appreciate your addition of the KNN baseline and the experiment with varying the universe size, as well as the clarifications on the training process. You could also consider mentioning in the related work that other model routing approaches do exist, but their code isn't publicly available and/or they haven't been peer reviewed-- otherwise, you might keep encountering the issue of readers wondering why it wasn't possible to include more routing works in the evaluations. I hope that you will include this new information in the revised paper.
>
> This new information increases my score to 5-- I still think the paper could use a bit more time to mature, but the evaluations are stronger now.

---

> > ### Author Response · Authors · 2024-11-26
> > **Revision of the paper**
> >
> > Thank you for your constructive feedback. Based on your suggestions, we have revised our publication to address the following points:
> >
> > - **Itemized Cost Table**: We have added the cost associated with data generation in Appendix C for clarity. Inference costs are already presented in Table 1. As the training was performed using the LoRA method, its cost is negligible.
> >
> > - **KNN Baseline**: Due to space constraints in the main text, the KNN baseline is now comprehensively defined in Appendix B. Additionally, the results for KNN (open-source) and KNN (mix) are provided in Table 1 and discussed in Section 4.3.
> >
> > - **Experiment with Varying Universe Size**: We have included further analysis of this experiment in Appendix D, due to space limitation.
> >
> > - **Related Work on Model Routing Approaches**: Section 2 has been updated to include additional relevant prior work and clarify how our approach is significantly different and novel compared to previous work. Section 4.1 has also been revised to explain why some prior works were excluded from comparison
> >
> > Could you please review these revisions and update your evaluation score accordingly?

---

> > > ### Author Response · Authors · 2024-11-27
> > > **Reminder to Reviewer NdR4**
> > >
> > > We would greatly appreciate it if you could review the revised version and update your evaluation score accordingly. Thanks.

---

> > > > ### Comment · Reviewer_NdR4 · 2024-11-27
> > > >
> > > > I appreciate your updating the paper. I updated my score from 3 to 5 this week, taking into account the new information that has been added. Thanks again for responding to my feedback and suggestions!

---

> > > > > ### Author Response · Authors · 2024-11-27
> > > > > **Official Comment by Author**
> > > > >
> > > > > In our latest revision, we have addressed all your concerns and the identified weaknesses in the paper. Could you please clarify the reasons for maintaining the score as a 'rejection'?

---

> ### Author Response · Authors · 2024-11-28
> **Official Comment by Author**
>
> To better understand your reasoning and align the review with the proposed evaluation score, I will summarize your concerns regarding the weaknesses and explain how we **completely. addressed** them:
>
> > W1. The experimental results are promising but a little bit sparse right now...
>
> We have added additional KNN strong baseline and all requested experiments. Here are the references:
> - **KNN baseline**: [link](https://openreview.net/forum?id=RQ9fQLEajC&noteId=G7glzBX3Bw).
> - **Revision and added experiments**: [link](https://openreview.net/forum?id=RQ9fQLEajC&noteId=FVdwQo4Nyv), revision reference: [link](https://openreview.net/forum?id=RQ9fQLEajC&noteId=EI3X0CYRCg), as **you suggested**
>
> > W2. It seems challenging to extend this method beyond datasets like MMLU that have multiple-choice answers...
>
> The problem formulation and the methods proposed in Section 3 are **clearly applicable to a wide range of tasks, as they do not include any specific optimizations for the MMLU benchmark**. The MMLU benchmark is used as an example for evaluation, but the proposed model has not been specifically adapted for this particular task.
>
> > W3. This equation uniquely takes into account the model performance, and does not consider model cost....
>
> Our experiment **aims to achieve the highest quality while minimizing inference costs**, so we optimize the set of underlying models for maximum performance. Future research can build on this to include other constraints, including inference cost.
>
> > W4. “We could not find any publicly available trained models for routing or model integration”...
>
> As noted in the [paper revision](https://openreview.net/forum?id=RQ9fQLEajC&noteId=EI3X0CYRCg), **we thoroughly examined previous work and explained why they were excluded from direct comparison**. Please refer to Section 4.1 and Footnote 6 for details. Furthermore, we added a **strong KNN baseline** that achieves competitive results with models like LLaMA2 13B, Mistral 7B, and LLaMA2 7B, and even shows competitiveness with LLaMA2 70B. This establishes it as a solid baseline for comparison.
>
> > W5. I don’t understand the claim in the Introduction (and restated at the very end of evaluation)...
>
> Our goal was to inspire the research community to create new LLMs by leveraging the knowledge from existing models, rather than pre-training entirely from scratch, which is much more expensive. Since MoE-style LLMs was emerging as a new approach, we reference and compare against them. Unlike MoE, which uses gating to select among various expert sub-networks within each layer for predicting the next token, our approach routes the entire query to the best underlying expert. This comparison highlights the difference: **MoE focuses on token-wise routing among experts, whereas our method, Routoo, performs query-wise routing among experts**.

---

> > ### Author Response · Authors · 2024-12-02
> > **Official Comment by Author**
> >
> > We kindly ask you to review the response above and consider reflecting it in your rating, as we believe we have effectively addressed your concerns and the proposed weaknesses (W1 to W5) outlined in the main review.

---

> > > ### Author Response · Authors · 2024-12-04
> > > **Official Comment by Author**
> > >
> > > We kindly request to review the response [above](https://openreview.net/forum?id=RQ9fQLEajC&noteId=0hlEBI3JRz) and consider updating your rating. We believe we have thoroughly addressed your concerns and the identified weaknesses (W1 to W5) highlighted in your main review.
> > >
> > > Here is the link to the revision that addressed your concerns: [thread](https://openreview.net/forum?id=RQ9fQLEajC&noteId=EI3X0CYRCg).

---

### Meta-Review · Area_Chair_ZTxy · 2024-12-20

**Metareview:**

This paper proposes a routing framework, Routoo, to balance the inference ost and quality. It comprises a performance predictor and cost-aware selector. Experiments show Routoo matches the performance of Mixtral 8x7b while reducing inference cost, and also nearly matches GPT4 with less cost.

Pros:
- Well written, and a clear description of the performance-cost trade-off in the routing framework.
- Competitive performance with GPT4 at reduced cost on the MMLU benchmark

Cons:
- Unclear novelty and lack of comparison with prior works
- Limited empirical evaluation on the MMLU benchmark

**Additional Comments On Reviewer Discussion:**

There are extensive discussion between authors and reviewers during the rebuttal phase. Although the authors addressed a few concerns around the cost of the routing algorithms and reviewers raise their ratings accordingly, the majority of reviewers still suggest a rejection after the change of ratings. The remaining concerns that are not well resolved include the comparison with prior works, and insufficient evaluation on broader benchmarks.

---

### Decision · Program_Chairs · 2025-01-22

Reject